# Automatically Processing IFC Clipping Representation for BIM and GIS Integration at the Process Level

**Junxiang Zhu [1], Peng Wu [1] , Mengcheng Chen [2], Mi Jeong Kim [3], Xiangyu Wang [4,5,\*] and Tingchen Fang [6]**

1. School of Design and the Built Environment, Curtin University, Bentley, WA 6102, Australia; junxiang.zhu@curtin.edu.au (J.Z.); peng.wu@curtin.edu.au (P.W.)
2. School of Civil Engineering and Architecture, East China Jiaotong University, Nanchang 330013, China; mcchen@ecjtu.edu.cn
3. School of Architecture, Hanyang University, Seoul 04763, Korea; mijeongkim@hanyang.ac.kr
4. College of Civil Engineering, Tongji University, Shanghai 200092, China
5. Australasian Joint Research Centre for Building Information Modelling, Curtin University, Bentley, WA 6102, Australia
6. Shanghai Construction Group Co. LTD, Shanghai 200080, China; fangtc@scgtc.com.cn
* Correspondence: xiangyu.wang@curtin.edu.au; Tel.: +61-041-668-3029

**Abstract:** The integration of building information modeling (BIM) and geographic information system (GIS) is attracting more attention than ever due to its potential benefits for both the architecture, engineering, and construction (AEC) domain and the geospatial industry. The main challenge in BIM and GIS integrated application comes from the fundamental data conversion, especially for the geometric information. BIM and GIS use different modeling paradigms to represent objects. The BIM dataset takes, for example, Industry Foundation Classes (IFC) that use solid models, such as boundary representation (B-Rep), swept solid, constructive solid geometry (CSG), and clipping, while the GIS dataset mainly uses surface models or B-Rep. The fundamental data conversion between BIM and GIS is the foundation of BIM and GIS integrated application. However, the efficiency of data conversion has been greatly impaired by the human intervention needed, especially for the conversion of the clipping geometry. The goal of this study is to automate the conversion of IFC clipping representation into the shapefile format. A process-level approach was developed with an algorithm for instantiating unbounded half spaces using B-Rep. Four IFC models were used to validate the proposed method. The results show that (1) the proposed approach can successfully automate the conversion of IFC clipping representation into the shapefile format; and (2) increasing boundary size has no effect on the file size of unbounded half spaces, but slightly increases the producing time of half spaces and processing time of building components. The efficiency of this study can be further improved by using an open-source package, instead of using the low-efficiency packages provided by ArcGIS.

**Keywords:** building information modeling (BIM); geographic information system (GIS); geometry transformation; shapefile; Industry Foundation Classes (IFC); 3D

## 1. Introduction

The building information modeling (BIM) and the geographic information system (GIS) emerge from different areas as follows: GIS starts as a mapping tool and is currently a comprehensive geospatial data management and analysis platform used in the geospatial industry [1] and BIM is widely used

in the architecture, engineering, and construction (AEC) domain [2] and is a platform for building information creation, management, and sharing [3,4]. The integrated application of BIM and GIS has been applied in various fields and has benefited both areas. In the AEC domain, BIM and GIS have been jointly used in various stages of the building life cycle, such as planning, construction, operation, and maintenance. They have been applied to disassemble offshore platforms [5], build retrofit [6], green building design [7], construction site layout optimization [8], and construction supply chain management [9]. For the geospatial industry, they have been used to assess building-level flood damage [10] and room-level traffic noise [11], as well as improve indoor navigation [12,13].

The main challenge with BIM and GIS integration comes from the conversion of geometric information between BIM and GIS [14,15]. On the BIM side, the representative Industry Foundation Classes (IFC) standard uses multiple methods to represent three-dimensional (3D) objects, including boundary representation (B-Rep), swept solid, constructive solid geometry (CSG), clipping, mapped representation, and, in some cases, surface model [16]. Among them, B-Rep, CSG, and swept solid are classic methods for solid modeling [17], and clipping is considered to be a special type of CSG, as only the Boolean difference operation is involved. As a matter of fact, several studies simply referred to clipping as CSG [18,19]. However, IFC explicitly differentiates them from each other, as two distinctive classes have been defined, i.e., *IfcBooleanClippingResult* and *IfcCsgSolid*. A mapped representation is a representation that uses any of CSG, B-Rep, and swept solid representations of other entities, usually after coordinate transformation. Swept solid, clipping, and CSG representations are implicit models defined by certain parameters, the exact shapes are only calculated when needed, for example, for display, whereas B-Rep representations are explicit models [17]. Figure 1 represents examples of B-Rep, swept solid, CSG, and clipping. In contrast, on the GIS side, City Geography Markup Language (CityGML) mainly uses B-Rep and surface model. B-Rep is used to represent room and building spaces, whereas surface models are used to represent internal or external walls, ceilings, windows, doors, and roofs. Another frequently used data format on the GIS side is the shapefile format. Multipatch is one of its shape types that uses B-Rep to represent 3D objects [20].

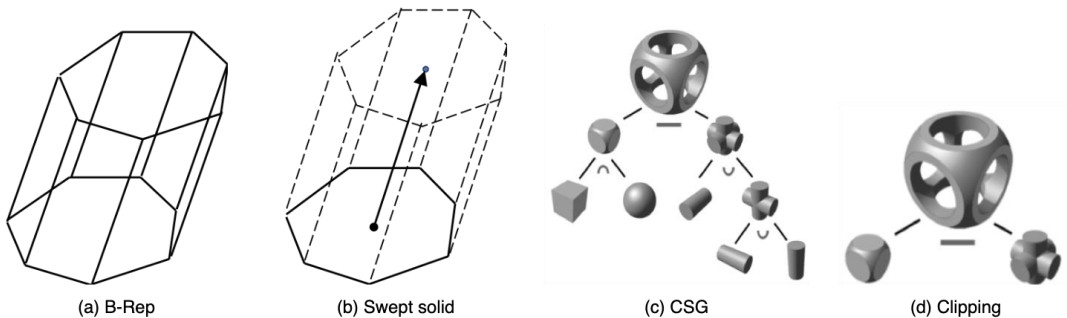

**Figure 1.** Basic representation methods used in IFC. (**a**) Boundary representation (B-Rep); (**b**) swept Solid; (**c**) constructive solid geometry (CSG) [21]; and (**d**) clipping.

Automated conversion of BIM datasets to GIS datasets is of great significance, given the fact that a great number of building models are needed for the construction of virtual city models that can be used in studies regarding smart city, etc., and because manual creation of such a large quantity of building models is time-consuming, cumbersome, and of low efficiency [18]. The automatic conversion has been realized for IFC B-Rep and swept solid [22], but not for IFC clipping. A clipping is the result of the Boolean difference between a swept solid and a half space solid, or between the result of the Boolean difference and a half space solid [23]. Clippings is classified into either one-clipping or multi-clipping depending on the number of half spaces involved. The primary challenge in automating IFC clipping conversion is the unbounded half space used by multi-clipping, because it is impossible to use bounded B-Rep to exactly represent boundless spaces. As a result, the conversion of clipping representations

can only be completed manually or semi-automatically [24], and in some cases, the use of clipping representations is avoided.

Therefore, the aim of this study is to automate the conversion of IFC clipping representation into a GIS shapefile dataset. Prior to that, a sub-objective is to develop a method to approximately represent half spaces using B-Rep. The remainder of this paper is organized as follows: Section 2 introduces the research background and related works; Section 3 explains the methodology of this study, including the detailed workflow for processing clipping representation and instantiating bounded and unbounded half spaces, as well as data used in this study; Section 4 uses building models to determine the appropriate boundary size for instantiating unbounded half spaces and validate the proposed method; Section 5 gives the discussion of this study; and Section 6 concludes this paper.

## 2. Research Background and Related Work

### 2.1. IFC-to-Shapefile Conversion

In general, there are two approaches for BIM data to be converted to GIS, including IFC-to-CityGML conversion and IFC-to-Shapefile conversion. IFC-to-CityGML conversion mainly utilizes semantic-based methods to transfer semantic information between IFC and CityGML by creating intermediate data models (ontologies) between IFC and CityGML [25]. IFC-to-Shapefile conversion is mainly used by studies that are focusing on the integrated application of BIM and GIS to solve practical problems [16]. The IFC-to-CityGML conversion faces more challenges and the challenges are even more severe as compared with IFC-to-Shapefile conversion. Apart from the common geometry conversion problems, two extra problems confronting IFC-to-CityGML conversion include mapping of level of details (LoDs) and converting solid models to surface models as follows: (1) CityGML has defined five LoDs from LoD0 to LoD4 [26] and BIMForum also defined five levels of development (LOD) for BIM [27]. Unfortunately, LoDs and LODs cannot be simplified and mapped in a one-to-one manner. (2) Conversion from solid models to surface models has to be properly completed. For example, a wall represented by swept solid in IFC has to be represented by two pieces of individual surfaces in CityGML, one for interior wall and the other for exterior wall. Such a conversion is more challenging than the conversion between solid models such as IFC-to-Shapefile conversion. The difference between a surface model and a solid model is shown in Figure 2, using a house model. In addition, the degree of challenge increases dramatically when converting IFC to higher CityGML LoDs. By far, IFC models can be automatically converted to valid LoD3 models [18].

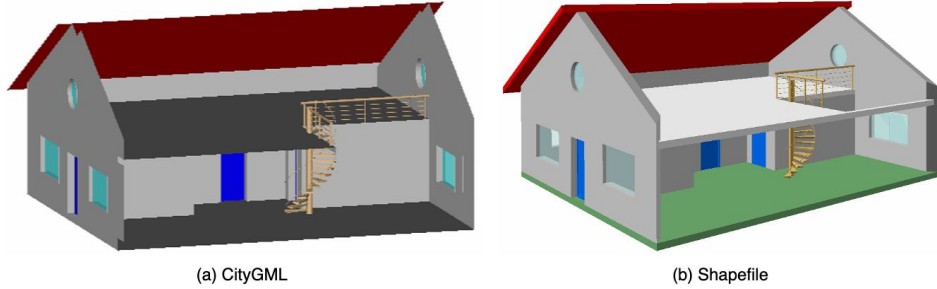

**Figure 2.** A house model in (**a**) CityGML and (**b**) Shapefile.

IFC-to-Shapefile conversion is theoretically easier to realize than IFC-to-CityGML conversion, as Shapefile uses solid models. In addition, it is more practical, as shapefile is a native format of ArcGIS, which is the most frequently involved GIS platform in integrated application of BIM and GIS [28], which means shapefile datasets can be directly used by ArcGIS while CityGML datasets need to be first converted.

## *2.2. Data Conversion from BIM to GIS*

Data conversion between BIM and GIS can be generally categorized into two groups: within-industry conversion and cross-industry conversion [24]. Within-industry conversion refers to data conversion that takes place between different applications within the same industry, for example, between different BIM systems. Such a conversion is easier, as similar data standards and conversion methods are usually being adopted by different applications in a specific industry. For example, most systems in BIM use IFC for data exchange [29]. Cross-industry conversion, as suggested by its title, refers to data conversion that takes place between different applications in different domains, for example, between BIM and GIS. A comparison with within-industry conversion shows that cross-industry conversion faces many more challenges, including: (1) conversion between different types of representations; (2) coordinate system transformation; and (3) other problems caused by the use of different receiving data formats, such as CityGML and shapefile. A good example that demonstrates the difference between within-industry and cross-industry conversion is the study conducted by Boyes et al. [30]. In their study, the geometry conversion was realized between MicroStation DGN file and the SketchUp file using Bentley BIM extensions (within-industry conversion), whereas problems arose unexpectedly in their attempts to convert IFC files into a shapefile format using Feature Manipulation Engine (FME).

While geometry conversion between different representation types, for example, CSG, B-Rep, and swept solid, has been well studied within the computer-aided design (CAD) and BIM area [31,32], it still remains a challenging problem for conversion between BIM and GIS [19,33–35], especially the conversion of clipping. In a study which aimed to implement building information models in geospatial context, Isikdag et al. developed methods for converting IFC data into shapefile and geodatabase formats [35]. However, this study had several limits, such as incorrect spatial location of transformed models, as well as a lengthy processing time and incompleteness in geometry conversion. In terms of geometry conversion, their methods did not support walls defined by four or more clippings and walls defined by three clippings could contain errors after conversion. In addition, the converted top faces of walls were presented as a set of unordered points in their study which impaired the usefulness of these models. In another study, although the geometry conversion problem was not pointed out explicitly [36], it could be inferred from the figures they presented, in the paper, that the walls represented using clipping were not successfully converted. In some studies, the conversion of clipping was even neglected. Deng et al. developed methods to convert IFC geometry, however, they failed to consider the clipping method [34]. Zhu et al. developed the open-source Approach (OSA) for converting IFC dataset into a shapefile dataset [24], however, the conversion process for clipping was not fully automatic and human intervention was needed.

Some commercial products, such as Data Interoperability Extension for ArcGIS (DIA) [37] and the aforementioned FME [38], also provide the possibility to convert IFC datasets to GIS datasets. However, these products are not sufficiently reliable for geometry conversion and can crash during operation [15]. Boyes et al. investigated this issue and found it was caused by the inability of FME in converting CSG and clipping to B-Rep [30]. Since DIA uses FME as the geometry conversion engine, it has the same problem with FME. All in all, currently, methods for converting clipping are not reliable or efficient.

## 3. Methodology

In this study, an automatic approach has been developed to achieve efficient data conversion from IFC clipping representations into a shapefile dataset, considering the aforementioned facts regarding data conversion between BIM and GIS that (1) commercial tools are not reliable enough, (2) current studies cannot automatically and efficiently convert IFC clipping representations to GIS dataset, and (3) cross-industry conversion faces many more challenges than within-industry conversion. The suggested automatic conversion process generally contains three steps: extracting, instantiating, and clipping.

### 3.1. Extracting Parameters for Clipping

Details about clipping representation in IFC is stored in the *IfcBooleanClippingResult* class. Each clipping entity has two components, i.e., a first operand and a second operand. The first operand can be a swept solid or another clipping entity. If the first operand is a swept solid, then the clipping is referred to as one-clipping in this study. Otherwise, if the first operand is another clipping, then, the clipping is referred to as multi-clipping. Thus, the clippings in a clipping representation, then, form a cascading system (see Figure 3). The number of clippings included in a clipping entity is referred to as clipping depth in this study. The second operand is a half space, which can be either a bounded half space or an unbounded half space.

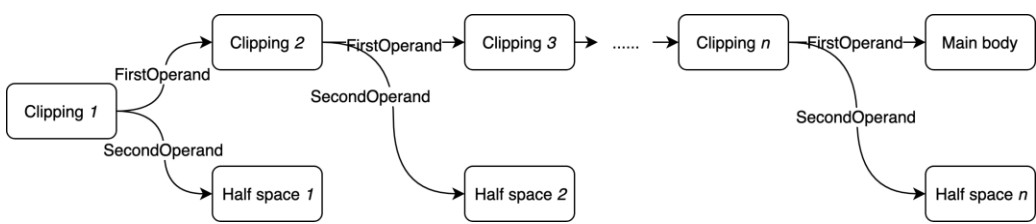

**Figure 3.** Structure of multi-clipping, i.e., the clipping chain.

In this study, the swept solid in the last clipping is referred to as the main body, while the second operands are referred to as clipping parts. From this point of view, one-clipping is a clipping entity with one main body and one clipping part, while multi-clipping has one main body and multiple clipping parts. Figure 3 presents the structure of a multi-clipping entity with a clipping depth of *n*.

Parameters for the main body include sweeping profile, sweeping direction, and sweeping depth; parameters for unbounded half space include a base surface (defined by a normal direction and a point on the surface) and an agreement flag (a Boolean value, either true or false). Since bounded half space (IfcPolygonalBoundedHalfSpace) is a subclass of unbounded half space (IfcHalfSpaceSolid), apart from the base surface and the agreement flag, it additionally includes two more parameters, including position and the polygonal boundary. The parameters for each type of half space are presented in Figure 4, which represents the relationship between unbounded half space and bounded half space using unified modeling language (UML) [39].

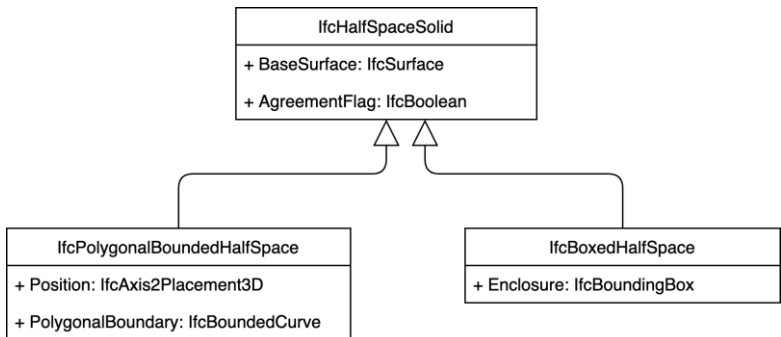

**Figure 4.** Unified modeling language (UML) for IfcHalfSpaceSolid, IfcPolygonalBoundedHalfSpace, and IfcBoxedHalfSpace [40].

The primary challenge in parameter extraction is to identify all the second operands (half spaces) and the first operand of the last clipping (the main body) from the cascading system. The following algorithm was developed to solve this problem (see Figure 5), and the pseudocode for this algorithm is given in Appendix A. This algorithm was used to process both multi-clipping entities and one-clipping entities.

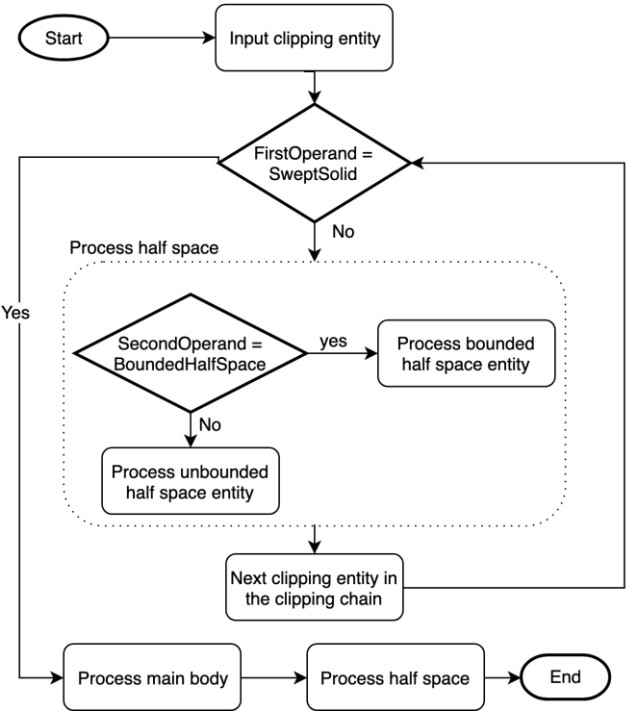

**Figure 5.** Algorithm for processing clipping entity.

In this algorithm, after inputting a clipping entity, the first step is to determine the type of clipping. If the first operand is a swept solid, then, the clipping is a one-clipping. In this case, only one main body and one clipping part should be retrieved and processed. If the first operand is another clipping entity, then, the initial clipping is a multi-clipping. In this case, the second operand is first retrieved and processed; then, the first operand of the next clipping in the clipping chain should be iteratively checked until the main body has been acquired. During this process, the second operands are iteratively retrieved and processed.

*3.2. Instantiating Bounded and Unbounded Spaces*

3.2.1. Challenges in Automating Clipping Conversion

Depending on the type of clippings and the type of half spaces involved in a clipping entity, there are four subtypes of clipping, i.e., (a) one-clipping with bounded half space, (b) one-clipping with unbounded half space, (c) multi-clipping with bounded half space, and (d) multi-clipping with unbounded half space. Given the fact that it is already possible to automatically convert one-clippings and multi-clippings with bounded half space [24], the main challenge of automating the conversion comes from multi-clipping with unbounded half space (see Table 1), mainly due to the fact that it is impossible to represent an infinite space (unbounded half space) using B-Rep that can only enclose a finite space.

**Table 1.** Types of clippings and types of half spaces.

|  | Automatic Conversion? | |
| --- | --- | --- |
|  | **Bounded Half Space** | **Unbounded Half Space** |
| One-clipping | Yes | Yes |
| Multi-clipping | Yes | No |

To solve this problem, this study suggested assigning an imagined boundary to unbounded half space to create a spatially sufficiently large B-Rep. Then, the challenge was simplified to determine the size of the imagined boundary. This was solved by analyzing a number of IFC models to obtain the maximum dimension of building components. A B-Rep was considered to be sufficiently large in space as long as it was larger than all building components in any dimension (*x*-axis, *y*-axis, and *z*-axis).

### 3.2.2. Procedure for Instantiation

The detailed steps for the proposed method of instantiating unbounded half space entities are presented Figure 6 as follows: (1) Retrieving the unbounded half space entity; (2) retrieving the cutting plane, including its normal direction, a point on it, and the agreement flag; (3) defining the size of the imagined boundary for unbounded half space, for example, 100 by 100 m; (4) getting the closed ring, *ring*, formed by the intersections of the cutting plane and lines that pass through vertices of the boundary and are perpendicular to the x-y plane; (5) obtaining the maximum and minimum value of the z-value, i.e., *max_z* and *min_z*, of the *ring*; (6) generating another two closed rings, i.e., *ring* 1 and *ring* 2. *Ring* 1, with a z-value of $(max\_z + \Delta)$, is parallel to the x-y plane; *ring* 2, with a z-value of $(min\_z - \Delta)$, is also parallel to the x-y plane. $\Delta$ is the length of a buffer that is created to make the B-Rep sufficiently large in the *z*-axis direction; (7) transforming the coordinates of all rings to the world coordinate system used by the IFC project; (8) determining the ring couple, i.e., (*ring* 1, *ring*) or (*ring*, *ring* 2) to be used to regenerate B-Rep, depending on the value of *AgreementFlag*; (9) generating B-Rep using OSA. This strategy can also be applied to convert bounded half space entities, all steps remain the same, except Step 3 where the boundary size is exactly retrieved from IFC. Please note that it is important to ensure that the converted B-Reps are closed, since only closed B-Reps can be used in geometry operation [17].

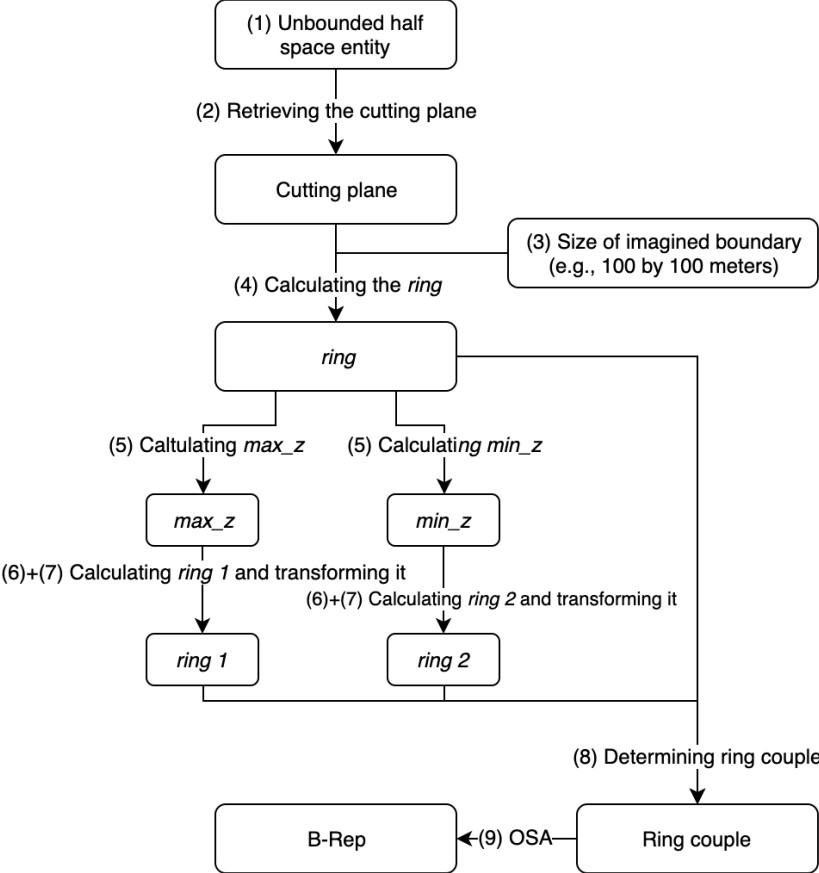

**Figure 6.** Steps for instantiating unbounded half space entities.

The equation for calculating the *ring* is as follows [24]:

$$\begin{bmatrix} X & Y & Z \end{bmatrix} = \begin{bmatrix} X & Y & AX_0 + BY_0 + CZ_0 \end{bmatrix} \times \left( \begin{bmatrix} 1 & 0 & 0 \\ 0 & 1 & 0 \\ A & B & C \end{bmatrix}^T \right)^{-1} \tag{1}$$

where $[X, Y, Z]$ are coordinates for the *ring*, $(A, B, C)$ is a normal direction to the cutting plane, $(X_0, Y_0, Z_0)$ is a point on the cutting plane, and $[X, Y]$ are coordinates for vertices of the imagined boundary. The boundary size in the x-y plane and the buffer in the z-axis direction can be determined after examining the size of building components. A B-Rep for an unbounded half space can be considered sufficiently large as long as the boundary size and the buffer is larger than the maximum size of building components. A number of BIM models are needed to examine the size of building components. In this study, square boundaries were used, and the size of a boundary was measured by its length, for example, a boundary with a size of 100 m means a square boundary with a dimension of 100 by 100 m.

Figure 7 provides an illustration of the imagined boundary for unbounded half space, as well as *ring*, *ring* 1, *ring* 2, and cutting plane. Four lines, from Line 1 to Line 4, are perpendicular to the x-y plane and go through the four corners of the imagined rectangular boundary, respectively. There are four intersections between these four lines and the cutting plane, which then form the *ring*. The maximum and minimum value of the z-value of these intersections are *max_z* and *min_z*, respectively. The coordinates for *ring* 1 and *ring* 2 can be obtained by adding or subtracting a length of buffer, which are $[X, Y, min\_z - \Delta]$ and $[X, Y, max\_z + \Delta]$, respectively.

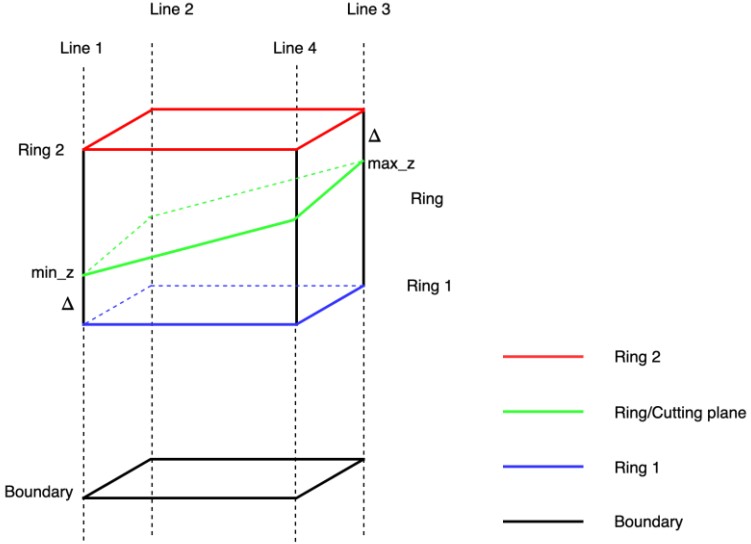

**Figure 7.** Illustration for boundary, ring 1, ring 2, ring, and the cutting plane.

### 3.2.3. Naming Rules for Clipping Components

To automatically identify the main body and corresponding clipping part(s) in the following step, an appropriate naming rule is needed for both main body and clipping parts during instantiation. The following naming rules were adopted in this study, as shown in Table 2. The main bodies are named with a prefix, "clipping_main", plus its unique identifier (ID); clipping parts are named with a prefix "clipping_part" plus its ID and a number, *n*, indicating its sequence in the clipping chain.

**Table 2.** Naming rules for main body and clipping part.

| Type | Name | Comments |
|------|------|----------|
| Main body | clipping_main_ID | ID: the unique identifier of the entity |
| Clipping part | clipping_part_ID_n | n: sequence in the clipping chain |

### 3.3. Clipping, Post-Processing, and Assessment

After obtaining B-Reps for the main body and all clipping parts (half spaces), the next task was to rebuild the B-Rep representations of the building components that were originally represented by clipping. Scripts based on ArcGIS were needed to automate this process. In general, three steps were involved as follows: (1) Acquiring all main bodies; (2) for each main body, acquiring all clipping parts; and (3) cutting the main body using clipping parts one by one in sequence. However, after this process, there were two issues caused by ArcGIS, which included: (1) Only geometric information and a unique identifier for each geometry are retained, while the original attached attributes get lost, and (2) merging multiple multipatch shapefiles is not currently supported by ArcGIS. As a result, additional scripts were needed to solve these issues to ensure data integrity and for better data management. The Python codes for solving these two issues are given in Appendix B. The developed method was assessed using criteria suggested by Donkers et al. [18], including model file size and processing time. Please note that the geometric uncertainty problem that is ubiquitous in mechanical computer-aided design (CAD) and computer-aided manufacturing (CAM), computational geometry, and many other fields [41] have been managed by using good quality IFC models which have defined an appropriate modeling precision.

### 3.4. Data

Several BIM models were used in this study, including two house models and two building models. Figure 8 presents these models, which include (a) House-1, (b) Institute, (c) House-2, and (d) Smiley. House-1, Institute, and Smiley were acquired from Karlsruhe Institute of Technology (KIT), which could be used unrestrictedly [42], while House-2 was acquired from Open IFC Model Repository [43]. House-1, Institute, and Smiley were used to measure the size of building components to determine the size of boundary and the buffer. House-2 was not used for this purpose as it failed to contain bounding box for each building part. During the validation of the proposed automatic conversion process, the Smiley model was not used because clipping was not used.

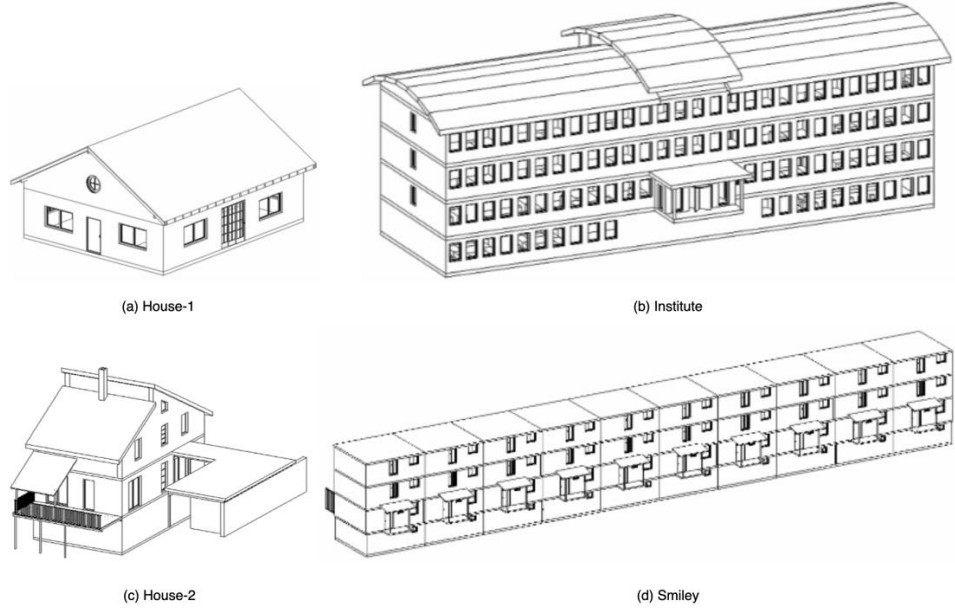

**Figure 8.** Building models used in this study.

Table 3 lists the building elements contained in each building model and their total quantities. It can be observed that various building elements, such as windows, doors, walls, and roofs are included in these models, and the quantity of entities for each model ranges from 82 to 831.

**Table 3.** Building elements in models and total quantity of entities for each element. "x" indicates the inclusion of a building element.

| Building Element | House-1 | Institute | House-2 | Smiley |
|---|---|---|---|---|
| IfcBeam | x | | x | x |
| IfcBuildingElementProxy | | | x | |
| IfcColumn | | x | x | x |
| IfcDoor | x | x | x | x |
| IfcMember | x | | | |
| IfcRailing | x | x | x | x |
| IfcRoof | | | x | |
| IfcSlab | x | x | x | x |
| IfcStair | x | x | x | x |
| IfcWall | x | | x | x |
| IfcWallStandardCase | x | x | x | x |
| IfcWindow | x | x | x | x |
| Total entity | 82 | 448 | 172 | 831 |

## 4. Validation and Results

### 4.1. Determing Bounday Size and the Buffer

To determine an appropriate size for the imagined boundary, the bounding boxes of all building elements are acquired and assessed. This is a reliable way to learn the size of building components, since in a well-built BIM model, each building component should have a bounding box representation. Three building models are used for this purpose.

Table 4 presents the result of the assessment. The maximum *XDim* for these models is between 7.5 and 42.0 m, the maximum *YDim* is between 7.3 and 18.0 m, whereas the number is between 3.6 and 4.2 m for the *ZDim*. For the boundary to be sufficiently large, its size should exceed the maximum *XDim*, *YDim*, and *ZDim* obtained in this test. In this study, the size of the boundary is, then, set to be 100 by 100 m, and the buffer, $\Delta$, is set to be 20 m.

**Table 4.** Maximum *XDim*, *YDim*, and *ZDim* for bounding boxes (B-Boxes) in House-1, Institute, and Smiley.

| | Quantity of B-Box | Maximum *XDim* | Maximum *YDim* | Maximum *ZDim* |
|---|---|---|---|---|
| House-1 | 82 | 13.0 m | 10.0 m | 3.6 m |
| Institute | 448 | 42.0 m | 18.0 m | 4.2 m |
| Smiley | 831 | 7.5 m | 7.3 m | 3.8 m |

### 4.2. Instantiating Half Spaces

With the obtained appropriate boundary size and buffer, the B-Reps for half spaces are automatically generated using the developed scripts. Figure 9 presents two examples of the generated B-Reps. To have a better view of them, two subfigures are provided for each half space representation with 0% and 30% transparency, respectively. In addition, the Institute model, colored by red, is placed along with them as a spatial reference. It is confirmed that the generated B-Reps for half spaces are sufficiently large in space to cover a building model.

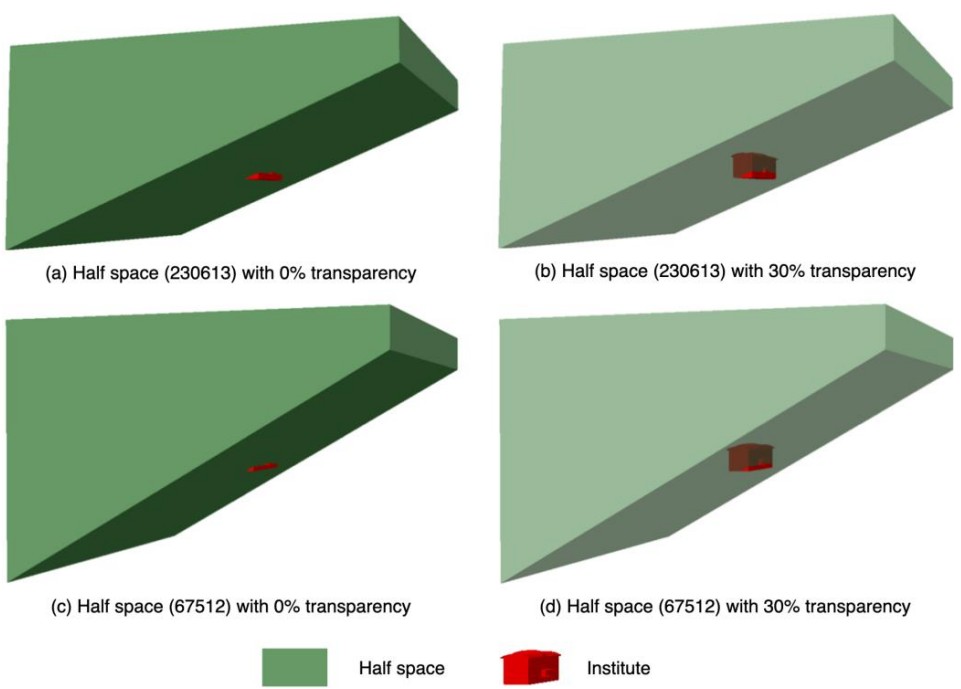

**Figure 9.** Examples of half spaces. (**a**) Half space (230613) with 0% transparency; (**b**) half space (230613) with 30% transparency; (**c**) half space (67512) with 0% transparency; and (**d**) half space (67512) with 30% transparency.

## 4.3. Clipping

Figure 10 presents an example of the clipping process using the wall entity with ID 228278 for the Institute model. The detailed clipping processes have been presented as well as the location of the wall in the final building model. The depth of this clipping is four, however, it is found that the first and second clipping is actually the same, thus, only three clippings are presented.

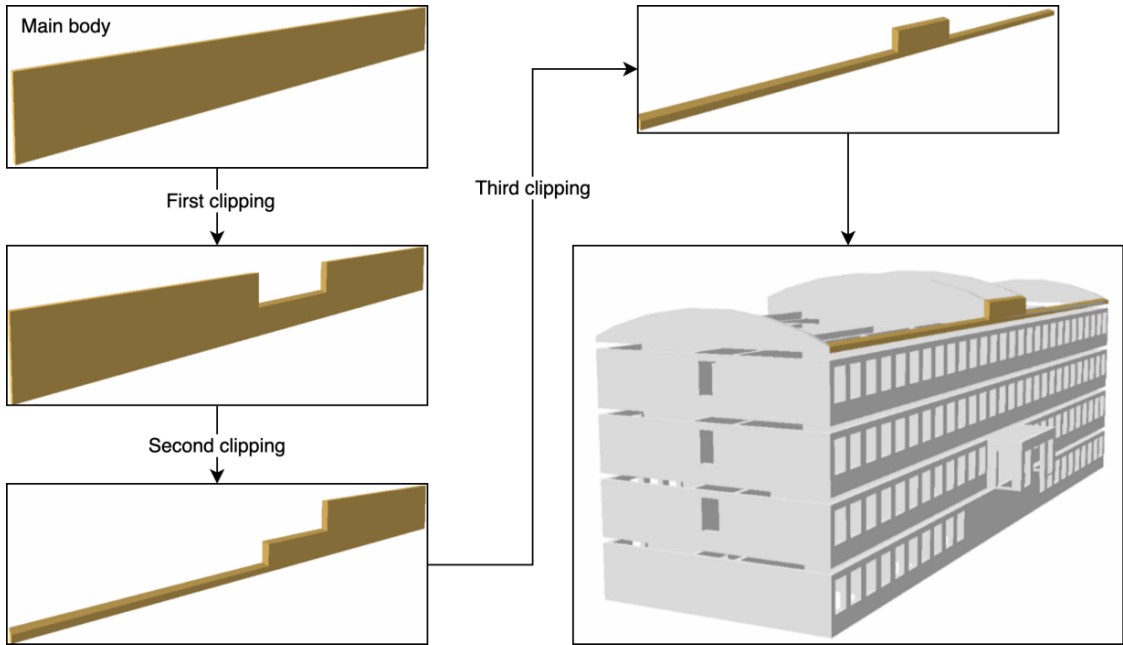

**Figure 10.** Generating a wall entity (ID 228278) for the Institute model.

Figure 11 represents the location of generated clipping geometry in the building models, i.e., House-1, Institute, and House-2 models. The geometry of these IFC models has been generated as expected. From the figure, it can be noticed that clipping is mainly used to represent irregular walls on the highest story of a building. Please note that the openings in the walls have not yet been extracted from the extruded walls.

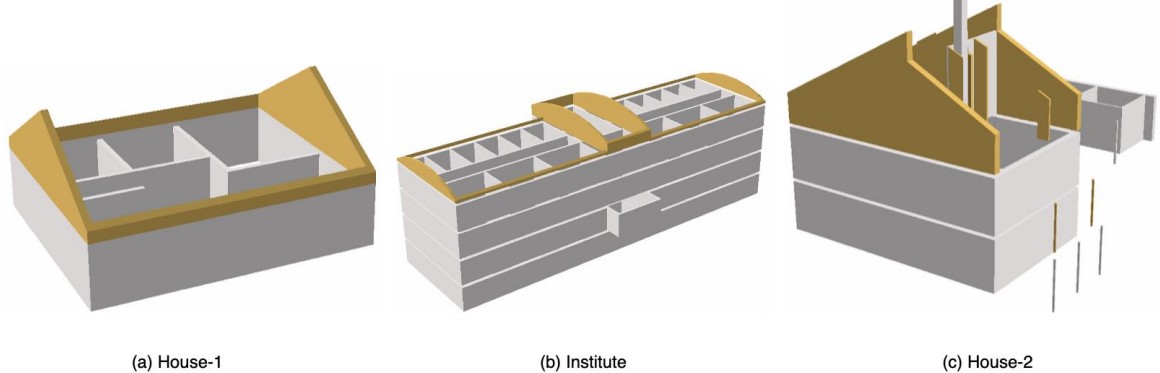

(a) House-1       (b) Institute       (c) House-2

**Figure 11.** Location of generated clipping geometry in each building model.

Details about building components originally represented by clipping are given below in Table 5, including entity ID, type of element, clipping depth, quantity of bounded half space and unbounded half space, and if the generated B-Rep is closed.

**Table 5.** Details about building components originally represented by clipping, including entity ID, type of element (Type), clipping depth (Depth), quantity of bounded half space (Bounded), quantity of unbounded half space (Unbounded), and if the generated B-Rep representation is closed (Is closed).

|          | Entity ID | Type   | Depth | Bounded | Unbounded | Is Closed? |
|----------|-----------|--------|-------|---------|-----------|------------|
| House-1  | 60012     | Wall   | 2     | 2       | 0         | Yes        |
|          | 67536     | Wall   | 1     | 0       | 1         | Yes        |
|          | 67828     | Wall   | 2     | 2       | 0         | Yes        |
|          | 75347     | Wall   | 1     | 0       | 1         | Yes        |
| Institute| 227944    | Wall   | 8     | 8       | 0         | Yes        |
|          | 228278    | Wall   | 4     | 4       | 0         | Yes        |
|          | 228563    | Wall   | 2     | 2       | 0         | Yes        |
|          | 228951    | Wall   | 6     | 6       | 0         | Yes        |
|          | 229235    | Wall   | 2     | 2       | 0         | Yes        |
|          | 229520    | Wall   | 2     | 2       | 0         | Yes        |
|          | 229966    | Wall   | 8     | 8       | 0         | Yes        |
|          | 230389    | Wall   | 7     | 7       | 0         | Yes        |
|          | 230647    | Wall   | 2     | 0       | 2         | Yes        |
|          | 230935    | Wall   | 2     | 2       | 0         | Yes        |
| House-2  | 36346     | Column | 1     | 0       | 1         | Yes        |
|          | 36396     | Column | 1     | 0       | 1         | Yes        |
|          | 102921    | Wall   | 1     | 1       | 0         | Yes        |
|          | 103158    | Wall   | 1     | 1       | 0         | Yes        |
|          | 103276    | Wall   | 1     | 1       | 0         | Yes        |
|          | 103338    | Wall   | 1     | 1       | 0         | Yes        |
|          | 103611    | Wall   | 1     | 1       | 0         | Yes        |
|          | 103724    | Wall   | 1     | 1       | 0         | Yes        |
|          | 102443    | Wall   | 2     | 2       | 0         | Yes        |
|          | 102583    | Wall   | 2     | 2       | 0         | Yes        |
|          | 102860    | Wall   | 1     | 1       | 0         | Yes        |
|          | 138980    | Wall   | 3     | 3       | 0         | Yes        |

For House-1, only four wall entities are represented by clipping with a maximum clipping depth of 2. For Institute, ten wall entities are represented by clipping with a maximum clipping depth of 8. For House-2, apart from ten wall entities, two column entities are also represented by clipping with a maximum clipping depth of 3. Both bounded and unbounded half spaces are involved, but the majority are bounded half spaces. As with House-1 and Institute, all the B-Reps are logically closed. For all three models, both bounded and unbounded half spaces are involved, and all the generated B-Rep representations have passed the "Is Closed" test, which means they are all logically closed solid model. This means the proposed method can automatically process any types of clipping, even those with a clipping depth larger than 4, and generate valid B-Rep into the shapefile format.

## 5. Discussion

### 5.1. Percentage of Clipping Entities

To have a general understanding of what methods are used to represent BIM models, a script has been written to calculate the number of entities that are represented by each representation method. Table 6 presents the number of entities for each building element that are represented by each representation method.

**Table 6.** Quantity of entities for each building element that are represented by B-Rep, swept solid, clipping, and surface.

| Model | IFC Class | B-Rep | Swept Solid | Clipping | Surface | Total |
|---|---|---|---|---|---|---|
| House-1 | IfcBeam | 3 | 1 | 0 | 0 | 4 |
| | IfcDoor | 5 | 0 | 0 | 0 | 5 |
| | IfcMember | 42 | 0 | 0 | 0 | 42 |
| | IfcRailing | 2 | 0 | 0 | 0 | 2 |
| | IfcSlab | 0 | 4 | 0 | 0 | 4 |
| | IfcStair | 1 | 0 | 0 | 0 | 1 |
| | IfcWallStandardCase | 0 | 9 | 4 | 0 | 13 |
| | IfcWindow | 11 | 0 | 0 | 0 | 11 |
| | Total | 64 | 14 | 4 | 0 | 82 |
| Institute | IfcColumn | 0 | 2 | 0 | 0 | 2 |
| | IfcDoor | 77 | 0 | 0 | 0 | 77 |
| | IfcRailing | 12 | 0 | 0 | 0 | 12 |
| | IfcSlab | 21 | 5 | 0 | 0 | 26 |
| | IfcStair | 0 | 0 | 0 | 4 | 4 |
| | IfcWallStandardCase | 0 | 111 | 10 | 0 | 121 |
| | IfcWindow | 206 | 0 | 0 | 0 | 206 |
| | Total | 316 | 118 | 10 | 4 | 448 |
| House-2 | IfcBeam | 0 | 39 | 0 | 0 | 39 |
| | IfcBuildingElementProxy | 0 | 0 | 0 | 8 | 8 |
| | IfcColumn | 0 | 8 | 2 | 0 | 10 |
| | IfcDoor | 14 | 0 | 0 | 0 | 14 |
| | IfcRailing | 6 | 0 | 0 | 0 | 6 |
| | IfcRoof | 1 | 0 | 0 | 0 | 1 |
| | IfcSlab | 0 | 8 | 0 | 0 | 8 |
| | IfcStair | 4 | 0 | 0 | 0 | 4 |
| | IfcWall | 11 | 36 | 10 | 0 | 57 |
| | IfcWallStandardCase | 0 | 36 | 10 | 0 | 46 |
| | IfcWindow | 25 | 0 | 0 | 0 | 25 |
| | Total | 61 | 91 | 12 | 8 | 172 |

It can be noticed that clipping is, in most cases, used to represent *IfcWallStandardCase* and sometimes *IfcColumn*. It is not a frequently used representation type, as the percentage of clipping

entities is only 4.9%, 2.2%, and 7.0% for the House-1, Institute, and House-2 model, respectively. The most frequently used representation types are B-Rep and swept solid.

## 5.2. Boundary Size, File Size, and Producing Time of B-Rep

To understand the relationship between boundary size, B-Rep file size, and time needed for the creation of B-Rep, a series of additional experiments have been conducted. B-Reps with boundary size of 10, 100, 1000, and 10,000 m for unbounded half spaces were first created, then their file size and time for creation were calculated.

Table 7 presents boundary sizes and file sizes of corresponding B-Reps for half spaces. The numbers clearly suggest that there is no relationship between them. B-Reps with a larger boundary size represent a larger space, but in terms of file size, they remain the same (1.16 KB). This relationship suggests that it is feasible to set the boundary size as large as possible, without the need to concern an over-large file size.

**Table 7.** Boundary sizes and file size of corresponding B-Reps for half spaces.

|  | Entity ID | 10 m | 100 m | 1000 m | 10000 m |
|---|---|---|---|---|---|
| House-1 | 67512 | 1.16 KB | 1.16 KB | 1.16 KB | 1.16 KB |
|  | 75323 | 1.16 KB | 1.16 KB | 1.16 KB | 1.16 KB |
| Institute | 230613 | 1.16 KB | 1.16 KB | 1.16 KB | 1.16 KB |
|  | 230623 | 1.16 KB | 1.16 KB | 1.16 KB | 1.16 KB |
| House-2 | 36317 | 1.16 KB | 1.16 KB | 1.16 KB | 1.16 KB |
|  | 36367 | 1.16 KB | 1.16 KB | 1.16 KB | 1.16 KB |

According to the initial experiments, the generation of B-Rep for a half space are completed in a short time, usually less than 1/1000 second. In this situation, the measured producing time can significantly deviate from the true value. It has been observed that, in each assessment, if the B-Rep generation process has only been run once, the time for each assessment varies significantly. Therefore, to make the assessment more reliable, the B-Rep generation operation has to be repeated multiple times in each assessment, which was set at 20,000 times in this study. Six unbounded half space entities are used for the assessment, and the total processing time is presented in Table 8. For each half space entity, it is observed that the time for each boundary size is very close, in most cases, within 0.5 s. Considering the large number of repetitions (20,000), the time difference between each execution is actually small enough that it can even be neglected.

**Table 8.** B-Rep producing time (20,000 repetitions) for unbounded half spaces.

|  | Entity ID | 10 m | 100 m | 1000 m | 10000 m | Max diff |
|---|---|---|---|---|---|---|
| House-1 | 67512 | 7.97 s | 8.04 s | 8.12 s | 8.14 s | 0.36 s |
|  | 75323 | 7.89 s | 7.68 s | 7.66 s | 7.55 s | 0.33 s |
| Institute | 230613 | 7.66 s | 7.56 s | 7.57 s | 7.68 s | 0.12 s |
|  | 230623 | 7.68 s | 7.84 s | 7.74 s | 7.66 s | 0.18 s |
| House-2 | 36317 | 8.85 s | 8.80 s | 8.17 s | 7.93 s | 0.91 s |
|  | 36367 | 8.54 s | 8.06 s | 7.96 s | 8.18 s | 0.58 s |

## 5.3. Boundary Size and Processing Time for Building Component

Because the boundary size of 10 m is too small to represent an unbounded half space, as shown in Table 4, then, it is not used in assessing the processing time of building components, instead, a larger boundary size, 100,000 m, has been added. For the same reason mentioned above, the processing time of 10 repetitions under the same condition is calculated. Five building components are used for

this purpose. The total processing time for each building component under each boundary size is presented in Table 9.

**Table 9.** Processing time (10 repetitions) for building components.

|  | Entity ID | 100 m | 1000 m | 10,000 m | 100,000 m | Max diff |
|---|---|---|---|---|---|---|
| House-1 | 67536 | 3.56 s | 3.60 s | 3.68 s | 3.76 s | 0.20 s |
|  | 75347 | 3.84 s | 3.92 s | 4.04 s | 4.17 s | 0.33 s |
| Institute | 230647 | 3.33 s | 3.46 s | 3.54 s | 3.71 s | 0.38 s |
| House-2 | 36346 | 3.53 s | 3.54 s | 3.77 s | 3.73 s | 0.24 s |
|  | 36396 | 3.96 s | 3.27 s | 4.09 s | 4.18 s | 0.91 s |

For building components from House-1 and Institute, there is a clear increasing trend in the processing time as the boundary size grows. The relationship between processing time and boundary size for these entities can be described using a linear equation, and high correlation coefficients have been noticed, around 0.98, as shown in Figure 12a. However, the increment in processing time is not very significant between neighboring boundary sizes. For example, consider the entity with ID 67536 from House-1 model, the total processing time increases to 3.76 s at the largest tested boundary size from 3.56 s at the smallest tested boundary size, with an increase of only 0.2 s, or 5.6%.

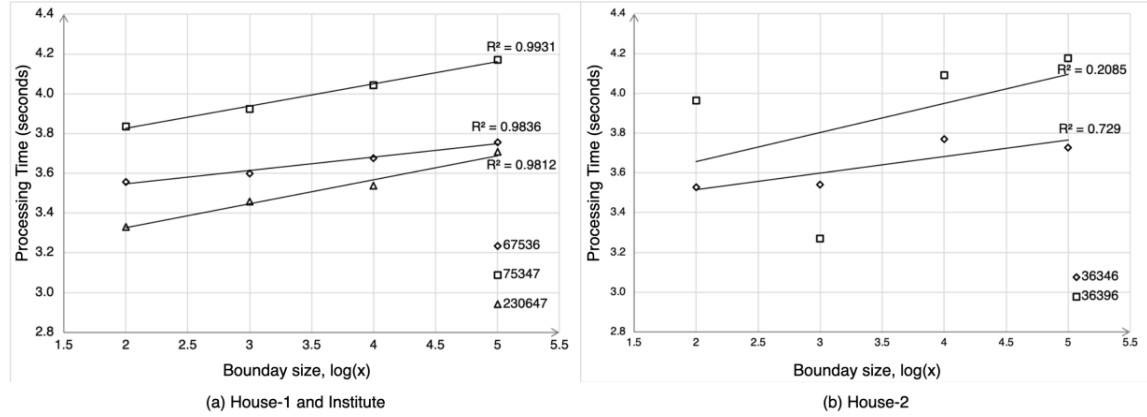

**Figure 12.** Linear regression analysis for (**a**) House-1 and Institute, and (**b**) House-2.

For the entities from House-2, however, the trend is not that obvious. Although trend lines can be drawn, they cannot accurately describe the data. As shown in Figure 12b, the correlation coefficients for entity ID 36346 and ID 36396 are only 0.729 and 0.208, respectively. However, they roughly show the positive correlation between processing time and boundary size. Considering that clipping is mainly used to represent walls and the usage is quite low, it is reasonable to believe that the processing time for the whole BIM model is not greatly affected by the increase of boundary size.

Please note that more boundary sizes need to be tested if a more accurate quantitative relationship between boundary size and processing time is to be determined, but this is not carried out in this study, as the quantitative relationship is not that decisive for this study.

### 5.4. Contribution to Data Conversion from BIM to GIS

This study processed all types of clippings in a fully automatic manner, including those with high clipping depths, and generated valid closed B-Rep in a shapefile format as compared with previous studies that focused on IFC-to-Shapefile conversion, such as studies by Isikdag et al. [19] and Zhu et al. [22,24]. This study benefited both IFC-to-Shapefile and IFC-to-CityGML conversion as

follows: (1) First, this study enhances the IFC-to-Shapefile conversion by automating the conversion of clipping. As a result, data conversion from BIM to GIS is realized in an easier, more efficient and more reliable way. This could be beneficial to application-oriented BIM and GIS integration studies, which mainly use the shapefile format to store geometric and semantic information from BIM [28]. (2) Second, the IFC-to-CityGML conversion also benefits from this study in two ways. On the one hand, the general knowledge about the clipping representation revealed by this study and how relevant information can be extracted from IFC file is helpful, as interpreting IFC is the first step for data conversion from BIM to GIS. On the other hand, as suggested by Donkers et al. [18], the IFC-to-CityGML conversion requires implicit models such as swept solid and CSG to be first converted to explicit models such as B-Rep. The shapefile format could then facilitate these studies by acting as an intermediate medium between IFC and CityGML.

## 6. Conclusions

This study proposed an approach for automatically converting IFC clipping representation to the shapefile format for the use in GIS. To achieve this goal, this study proposed to instantiate unbounded half spaces by creating B-Rep that is sufficiently large in space and using an imagined boundary. Several tests were performed to determine an appropriate size for the imagined boundary, including measuring the size of building components, assessing the relationship between boundary size and file size of half spaces, as well as the relationship between boundary size and producing time of half spaces and building components. Four BIM models, including two house models and two building models from KIT and Open IFC Model Repository were used to validate the proposed method. In comparison with previous studies that aimed to convert clipping geometry for use in GIS, this method is relatively easy to perform and can process all types of clippings, including multi-clipping with unbounded half space, and generate valid B-Rep into the shapefile format.

The main findings of this study include the following: (1) the proposed method can successfully automate the conversion of IFC clipping into the shapefile format, regardless the type of clipping and the type of half space; (2) increasing the boundary size will not increase the file size of corresponding B-Reps for half spaces, but it will slightly increase the producing time of half spaces and processing time of building components; and (3) based on the previous point, an appropriate boundary size has been suggested to instantiate unbounded half spaces using B-Rep, which is 100 by 100 m in the x-y plane and a buffer of 20 m in the *z*-axis direction.

The primary limitation of this work is that it currently depends on a Python site package provided by a commercial GIS product, ArcGIS. This site package is of low efficiency and can only run on the Windows system. In the future, an alternative open source package should be used to further improve data conversion efficiency and support more operating systems. Apart from the geometry conversion involved in this study, transferring semantic information from IFC into the shapefile format is another challenge that should be properly addressed in the future.

**Author Contributions:** Conceptualization, J.Z., P.W., and X.W.; Formal analysis, J.Z.; Funding acquisition, P.W. and X.W.; Methodology, J.Z.; Software, J.Z.; Writing—original draft, J.Z. and P.W.; Writing—review and editing, P.W., M.C., M.J.K., and T.F. All authors have read and agreed to the published version of the manuscript.

**Funding:** This research was supported by the Australian Government through the Australian Research Council's Discovery Project, grant number #DP180104026 and #DP170104613, and the Shanghai Economic and Information Commission Special Fund Programs, grant number Shanghai J-2018-27.

**Acknowledgments:** The authors would like to thank the anonymous reviewers for their comments and suggestions that helped improve the comprehensiveness and clarity of our paper.

**Conflicts of Interest:** The authors declare no conflict of interest.

## Appendix A

The pseudocode for processing clipping entity.

```
1:   procedure process-clipping(clipping):
2:       while the first operand of clipping is not a swept solid:
3:           if the second operand of clipping is a bounded half space entity:
4:               process bounded half space entity
5:           elseif the second operand of clipping is an unbounded half space entity:
6:               process unbounded half space entity
7:           end if
8:           clipping ← the first operand of clipping
9:       end while
10:      process main body entity
11:      if the second operand of clipping is a bounded half space entity:
12:          process bounded half space entity
13:      elseif the second operand of clipping is an unbounded half space entity:
14:          process unbounded half space entity
15:      end if
16:  end procedure
```

## Appendix B

Python codes for merging multipatches is given below. The code uses two custom functions for listing shapefile and initiating the shapefile writer, i.e., *listShapefiles* and *initiateShpWriter*.

```
1:   def combineMultipatches(workspace, outFeature):
2:       multipatches = listShpfiles(workspace)#list all shapefiles
3:       initial_shp = shapefile.Reader(multipatches [0])
4:       attList = [initial_shp.fields[i][0] for i in range(1,len(initial_shp.fields))]
5:       shpWriter = initiateShpWriter(attList) #initiate the shapefile writer
6:       for multipatch in multipatches:
7:           sf_multipatch = shapefile.Reader(multipatch)
8:           shapes = sf_multipatch.shapes()
9:           records = sf_multipatch.records()
10:          for i in range(len(shapes)):
11:              shape = shapes[i]
12:              record = records[i]
13:              points = np.hstack((np.mat(shape.points),np.mat(shape.z).T))
14:              points = points.tolist()
15:              parts = []
16:              for k in range(len(shape.parts)):
17:                  if k != len(shape.parts)-1:
18:                      ring = points[shape.parts[k]:shape.parts[k+1]]
19:                  else:
20:                      ring = points[shape.parts[k]:]
21:                  parts.append(ring)
22:              partTypes = (np.ones(len(parts))*5).tolist()
23:              shpWriter.poly(parts = parts, partTypes = partTypes, shapeType = 31)
24:              shpWriter.records.append(record)
25:      shpWriter.save(outFeature)
```

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
