# Peer review of "Automatically Processing IFC Clipping Representation for BIM and GIS Integration at the Process Level"

_applsci, doi:10.3390/app10062009_

Round 1

Reviewer 1 Report

Very good work, with a clear presentation of methods and results.

It is only missing to describe the mainstream of the article for designers and planners within the industry and how could the approach be useful for other research areas.

Author Response

Point 1. It is only missing to describe the mainstream of the article for designers and planners within the industry and how could the approach be useful for other research areas.

Response 1: 

Thanks for the comment.

BIM and GIS can be jointly used in several stages in the building life cycle, including not only designing and planning, but also operation and maintenance.

This point has been clarified in line 46-50, supported by a series of reference.

“BIM and GIS have been jointly used in various stages of the building life cycle, such as planning, construction, operation and maintenance. They have been applied to disassembly of offshore platforms [5], building retrofit [6], green building design [7], construction site layout optimization [8] and construction supply chain management [9]”

We have also explained how this study can potentially benefit other research areas. For example, this study focuses on BIM/GIS integration, and its contribution to data conversion from BIM to GIS has been described in section 5.4. This study focuses on IFC-to-Shapefile conversion, but it can also benefit other data conversion research areas, e.g. IFC-to-CityGML conversion, as pointed out in line 425-432:

“(2) Second, the IFC-to-CityGML conversion can also benefit from this study in two ways. On one hand, the general knowledge about the clipping representation revealed by this study and how relevant information can be extracted from IFC file can be helpful, as interpreting IFC is the first step for data conversion from BIM to GIS. On the other hand, as suggested by Donkers et al. [18], the IFC-to-CityGML conversion requires implicit models such as swept solid, CSG to be first converted to explicit models such as B-Rep. Shapefile can then facilitate these studies by acting as an intermediate medium between IFC and CityGML.”

Reviewer 2 Report

MDPI Applied Sciences 742042

Automatically processing IFC clipping representation for BIM/GIS integration at process level

Zhu et al.

The study proposed and developed an approach that automates the conversion of IFC clipping representation to shapefile for addressing existing challenges in GIS-BIM integration. The work has a paramount importance in geospatial, engineering, construction and asset management industries. I, therefore, think it will be worthy of eventual publication in Applied Sciences.

The article is a well-written and clear with detail background of previous endeavors, current challenges, and the methods followed and results achieved in this work. It is also free of grammatical and stylistic errors.  As, a result, I have very few specific comments below, suggesting further clarifications in the paper.

  1. One of the main challenges in the conversion process is loss of semantic and other information. And, the authors came up with Python codes for solving such issues. The conceptual approach employed in the codes to eliminate or minimize this should be made clear in the paper.
  2. The proposed integration can have geometric and topologic related uncertainties. Is there any way to evaluate these? It would be useful to include in the paper a sentence or two why you didn’t consider such uncertainties to evaluate the quality of the proposed integration.
  3. The statistics shown in Figure 12 raises the question of whether such correlations are meaningful with only four data points for each entity. Also, it raises a question of whether the selection of a wall (for house-1 and the institute shown in figure 12a) or column (for house-2, figure 12b) matters.  
  4. It would be helpful to show the number labels of selected clipping entities in figure 11.
  5. The authors can further improve the discussion section by incorporating some relevant literature.

Author Response

Point 1: 

One of the main challenges in the conversion process is loss of semantic and other information. And, the authors came up with Python codes for solving such issues. The conceptual approach employed in the codes to eliminate or minimize this should be made clear in the paper.

Response 1:

Thanks for the suggestion.

Loss of semantic information is a common problem during data conversion from BIM to GIS due to the incompatibility in data models/schemas used by these two areas. However, the “semantic loss problem” mentioned in this study was a different case, which was caused by ArcGIS, the data processing tool used in this study. After geometric operations within ArcGIS, such as difference and intersect, ArcGIS will dump the original attributes, and only keep the processed geometric model. As a result, Python codes were developed to reconnect the processed geometric model with the original attributes. Since this problem was caused by a specific tool, it is not common, that important to this study, and should not be a part of the methodology. That is why the Python codes were put in appendix for reference only.

To eliminate the possible misunderstanding, the statement that “there would be two issues after this process, including (1) only geometric information and a unique identifier for each geometry are retained, while all other attributes or semantic information get lost ……” has been revised as shown in lines 275-277:

“However, there would be two issues after this process caused by ArcGIS, including (1) only geometric information and a unique identifier for each geometry are retained, while the original attached attributes get lost, and ……”

Point 2

The proposed integration can have geometric and topologic related uncertainties. Is there any way to evaluate these? It would be useful to include in the paper a sentence or two why you didn’t consider such uncertainties to evaluate the quality of the proposed integration.

Response 2:

Thanks for the comment.

Geometric uncertainty is ubiquitous in mechanical CAD/CAM, computational geometry, and many other fields. Many models have been developed for handling imprecise geometry, such as the Linear Parametric geometric Uncertainty Model. However, in this study, this problem does not have to be concerned. According to prior experience, the geometric uncertainty would be a real problem when the modelling precision was not defined properly, and the transformed geometric model may have defects, such as having unwanted “parts” that should not have been there. However, as long as an appropriate modelling precision, say 1e-05, has been defined for IFC models, the conversion process proposed can be completed without problem. Therefore, the geometric uncertainty problem can be controlled in the beginning during the modelling phase. By controlling the quality of IFC models, the geometric uncertainty problem can be bypassed.

This point has been clarified in lines 282-285:

“Please note that the geometric uncertainty problem that is ubiquitous in mechanical computer-aided design (CAD)/computer-aided manufacturing (CAM), computational geometry, and many other fields [41] can be managed by using good quality IFC models which have defined an appropriate modeling precision.”

Point 3: 

The statistics shown in Figure 12 raises the question of whether such correlations are meaningful with only four data points for each entity. Also, it raises a question of whether the selection of a wall (for house-1 and the institute shown in figure 12a) or column (for house-2, figure 12b) matters.

Response 3:

Thanks for the comments.

Figure 12 is used to visually show the rough relationship between boundary size and processing time, which has already been revealed by Table 9. Figure 12 is additional supporting material for Table 9. More points would be needed if a more accurate quantitative relationship between boundary size and processing time is to be investigated, but this is not the focus of this study, nor that decisive to this study.

This point has been clarified in line 414-416:

“Please note that more boundary sizes need to be tested if a more accurate quantitative relationship between boundary size and processing time is to be determined, but this is not carried out in this study, as the quantitative relationship is not that decisive to this study.

As for the second question, there is no selection problem, as Figure 12 has included all the five entities that involve unbounded half space, including 67536 and 75347 for House-1, 230647 for Institute, 36346 and 36396 for House-2.

Point 4

It would be helpful to show the number labels of selected clipping entities in figure 11.

Response 4:

Thanks for the useful suggestion.

A problem with displaying the number labels in the figure is that the readability of the figure would be poor, considering the size of the figure and the number of labels (26).

We have adopted a strategy by using a different colour to differentiate the selected clipping entities from others.

Point 5:

The authors can further improve the discussion section by incorporating some relevant literature.

Response 5:

Thanks for the suggestion.

Several references have been added to support the discussion on the contribution of this study, including:

Zhu, J.; Wang, X.; Wang, P.; Wu, Z.; Kim, M.J. Integration of bim and gis: Geometry from ifc to shapefile using open-source technology. Automation in Construction 2019, 102.

Zhu, J.; Wang, X.; Chen, M.; Wu, P.; Kim, M.J. Integration of bim and gis: Ifc geometry transformation to shapefile using enhanced open-source approach. Automation in Construction 2019, 106.

Ma, Z.; Ren, Y. Integrated application of bim and gis: An overview. Procedia Engineering 2017, 196, 1072-1079.

Donkers, S.; Ledoux, H.; Zhao, J.; Stoter, J. Automatic conversion of ifc datasets to geometrically and semantically correct citygml lod3 buildings. Transactions in GIS 2016, 20, 547-569.

Isikdag, U. Towards the implementation of building information models in geospatial context. University of Salford, 2006.

See section 5.4:

“Compared with previous studies focusing on IFC-to-Shapefile conversion, such as studies by Isikdag et al. [18] and Zhu et al. [21,23], this study can process all types of clippings in a fully automatic manner, including those with high clipping depths ……”